# Plant-Wear: A Multi-Sensor Plant Wearable Platform for Growth and Microclimate Monitoring

**DOI:** 10.3390/s23010549

**Published:** 2023-01-03

**Authors:** Joshua Di Tocco, Daniela Lo Presti, Carlo Massaroni, Stefano Cinti, Sara Cimini, Laura De Gara, Emiliano Schena

**Affiliations:** 1Departmental Faculty of Engineering, Università Campus Bio-Medico di Roma, 00128 Rome, Italy; 2Department of Pharmacy, Università degli Studi di Napoli Federico II, 80138 Naples, Italy; 3Department of Science and Technology for Humans and the Environment, Università Campus Bio-Medico di Roma, 00128 Rome, Italy

**Keywords:** microclimate evaluation, plant wearable, precision agriculture, smart agriculture, strain sensor, wearable device

## Abstract

Wearable devices are widely spreading in various scenarios for monitoring different parameters related to human and recently plant health. In the context of precision agriculture, wearables have proven to be a valuable alternative to traditional measurement methods for quantitatively monitoring plant development. This study proposed a multi-sensor wearable platform for monitoring the growth of plant organs (i.e., stem and fruit) and microclimate (i.e., environmental temperature—T and relative humidity—RH). The platform consists of a custom flexible strain sensor for monitoring growth when mounted on a plant and a commercial sensing unit for monitoring T and RH values of the plant surrounding. A different shape was conferred to the strain sensor according to the plant organs to be engineered. A dumbbell shape was chosen for the stem while a ring shape for the fruit. A metrological characterization was carried out to investigate the strain sensitivity of the proposed flexible sensors and then preliminary tests were performed in both indoor and outdoor scenarios to assess the platform performance. The promising results suggest that the proposed system can be considered one of the first attempts to design wearable and portable systems tailored to the specific plant organ with the potential to be used for future applications in the coming era of digital farms and precision agriculture.

## 1. Introduction

Plants are a key element for a planet’s life due to their involvement in the carbon cycle being the primary producers in the food web. They are also responsible for enduring atmosphere composition and water balance. The increasing worldwide population entails more effort in guaranteeing food security. For this reason, agriculture must improve its performance and harvest capacity along with reducing the use of pesticides and agronomic procedures which negatively affect the environment and the harvest products’ quality [1].

In the context of sustainable agriculture, the optimization of the harvest may be supported by monitoring and quantifying plant growth and the influence of stressor factors. Plant stresses can be categorized into two classes: abiotic and biotic factors. Abiotic factors are related to the microclimate and include as an example, temperature level, irradiance, water availability, salinity, atmospheric carbon dioxide (CO_2_) enrichment, while biotic factors refer to damage caused by pest and pathogens [2,3,4]. Precision agriculture is a modern farming management concept that uses digital techniques to adjust and fine-tune land for optimizing agricultural production processes. Here, the key point is the management optimization. Smart agriculture is a more recent concept that investigates the use of innovative technology to improve agricultural production while at the same time lowering the inputs significantly. Here, the focus is rather on access to data and the application of these data. Hence, smart agriculture runs on the principles of precision agriculture shifting to a holistic and a more rounded approach where the focus is not only on management optimization but on the employment of smartest treatments. The use of these quantitative data to apply measures that are economically and ecologically meaningful makes precision agriculture sustainable. In the context of sustainable agriculture, the use of specific sensors for continuous monitoring of many parameters (e.g., microclimate, pesticides concentration, plant growth, soil properties, and illuminance) is becoming a common trend [4,5,6,7,8,9,10]. Among others, microclimate and growth are two of the main parameters that can provide useful information on plant wellbeing and can define the plant’s needs in terms of agronomic procedures. Indeed, monitoring growth and microclimate by employing wearable sensors close to the plants allows for improving the plants’ environmental conditions. Since the mentioned parameters can vary from side to side of a greenhouse or field, the use of wearables in different sites may be beneficial to retrieve a complete overview of the crop development. In addition, wearable technology provides an accurate spatial awareness of the plants’ health parameters with reduced bulkiness and weight [5,11].

To date, traditional technologies employed for plant health monitoring include contactless methods such as spectroscopy, machine vision systems, imaging techniques, and drones [12,13,14,15]. The use of remote systems faces some partial issues that are dampening their application in long-term plant monitoring. The lack of high spatial and temporal resolution and the low measurement reliability associated with these methods make them inadequate for the continuous tracking of plant organs development. Recently, new techniques have emerged for monitoring plant growth. Wearable systems embedding flexible strain sensors are reaching growing attention to overcome these issues thanks to their high stretchability and adaptability to plant organs. Most are conductive materials integrated within a polymer substrate or directly brushed on the plant surface with different principles of work [5,10,16,17]. Usually, a change in the electrical resistance or capacitance is experienced by the proposed sensing element under the growth of the engineered plant organs (e.g., stem, leaves, or fruit Recently, fiber optic sensors (i.e., fiber Bragg gratings)) have also been proposed for the same aims with interesting results [9,18].

Microclimate monitoring has been performed by employing Polydimethylsiloxane- (PDMS) encapsulated polyimide (PI) sensors [5,19], PI-based capacitive sensors and copper-based thermistor [11], functionalized FBG [9,18], and commercial all-in-one integrated circuits based on thermocouples and capacitive sensors [20].

Some of the literature available studies have proposed multi-sensor platforms for monitoring plant growth and the surrounding microclimate simultaneously. However, most of these are characterized by a high-priced and sophisticated fabrication process in conjunction with, in some cases, high-cost of read out electronics, and uneasy customizability [21,22,23,24,25].

In the present work, we present an easy-to-produce, low cost and unobtrusive multi-sensor platform based on conductive and environmental sensors for continuous, simultaneous, and wireless real-time monitoring of plant growth (i.e., stem and fruit) along with the plant’s microclimate. In particular, we present the sensors manufacturing, metrological characterization to strain and relative humidity—RH, and the experimental tests performed in a laboratory and open field settings on a tobacco and melon plant, respectively.

## 2. Materials and Methods

In the present section, we present the engineered plant, the design and development of the multi-sensor platform for monitoring plant growth (both in terms of stem growth and fruit growth) and its surrounding microclimate, the experimental tests, and the main steps of data analysis.

### 2.1. Plant Material

The proposed platform has been tested on two different plants. A tobacco plant (*Nicotiana tabacum*) was used to focus on stem growth in laboratory settings. In this controlled experiment, we sterilized the tobacco seeds by soaking in 70% ethanol for 2 min, followed by soaking in 5% bleach for 15 min. The seeds were then rinsed five times with sterilized water. Sterilized tobacco seeds were sowed in soil at 25 ± 1 °C with a photoperiod of 16 h light/8 h dark. A melon plant (*Cucumis melo*) was used to assess the feasibility of the proposed sensor for monitoring the fruit growth in an open field scenario.

### 2.2. Multi-Sensor Growth Monitoring Platform

The proposed sensors (for stem and fruit growth) represent the custom elements of a multi-sensor platform consisting of strain sensors for monitoring stem elongation and fruit expansion and a commercial environmental system (BME280 by BOSCH, Gerlingen, Germany) for measuring environmental T and RH.

#### 2.2.1. Stem Growth Sensor

The sensing element was obtained from a conductive textile sheet (Eeontex LG-SLPA by Eeonyx, Pinole, CA, USA) by hand cutting a custom shape of dimensions 60 mm × 10 mm (length × width). These dimensions have been chosen based on the plant sample available for testing. This type of sensor changes its resistance according to the applied strain. To connect the sensor to the electronics, conductive silver-plated threads (78/18 z turns HC + B by Shieldex, Bremen, Germany) have been sewn at the shorter ends of the sensor. Figure 1 shows a graphical representation of the sensor shape and its manufacturing process.

To transduce the resistance change of the sensor into a voltage, a Wheatstone bridge in 1/4 bridge configuration was used. The output of the Wheatstone bridge was sent to an instrumentation amplifier (AD8426, by analog devices, Wilmington, MA, USA) and digitized by a commercial board (M5 Stick-C Plus, by M5Stack, Shenzhen, China). To compensate for temperature (T) and relative humidity (RH) influence on the sensor’s output, another nominally identical sensor was positioned close to the plant but in the absence of strain (i.e., fixed on a rigid plastic bar) and its output has been collected using identical electronics as previously described. Data have been collected wirelessly using a Wi-Fi communication protocol on a Raspberry PI 4 (Raspberry Pi, Wales, UK). The microclimate parameters (i.e., T and RH) have been monitored using a commercial environmental sensor (BME280 by BOSCH, Gerlingen, Germany) connected via I2C to the M5Stick-C Plus.

#### 2.2.2. Fruit Growth Sensor

The sensing element was obtained using the same conductive textile described in the previous subsection. We chose different dimensions for the sensing element (i.e., 120 mm × 15 mm) according to the size of the fruit sample available for the open field experiments. Considering the specific scenario, we faced two main issues. The first one is related to the need for a custom fixing mechanism to allow the sensor to be placed on the fruit without permanent fixing. This issue has been addressed by providing the sensing element with two 11 mm in diameter metal snaps (Koh-i-Noor Hardmuth, Budějovice, Czechia). The male parts of the metal snaps have been fixed at the sensor’s extremities, and the female parts have been sewed at the extremities of an elastic textile strip (with lesser elasticity than the sensor in order to transfer the majority of strain to the sensor). Therefore, a sensing element that can be easily placed and removed around the fruit was obtained. Figure 1 shows a graphical representation of the sensor shape and its manufacturing process. The second issue regards the need for compensation or mitigation of T and RH influence on the sensor’s output since the open field application exposes it to relevant changes in these two environmental parameters. This concern has been addressed by manufacturing the sensor into a three-layer structure composed of a dual outer layer polymeric matrix surrounding the inner layer represented by the sensor. The three-layer structure has been obtained by multiple casting of a silicone polymer (Ecoflex 00-30, by Smooth-On, Inc., Macungie, PA, USA). Firstly, a casting mould was 3D-printed in polylactic acid (PLA), and the first silicone layer was cast in the mould. After ∼1 h curing time, the sensor cut from the sheet was positioned with the metal snaps at its extremities. Then, the top outer layer was cast to entirely cover the sensor. The total amount of silicone used was ∼8 g. Finally, after ∼8 h curing time, the final sensor was removed from the mould.

To transduce the resistance change of the sensor into a voltage, a Wheatstone bridge in half-bridge configuration was used. Another nominally identical sensor was placed on the adjacent branch of the bridge to compensate for any T or RH influence on the measurement. The output of the Wheatstone bridge was sent to an instrumentation amplifier (AD8426, by analog devices, Wilmington, MA, USA) and digitized by a commercial board (M5 Stick-C Plus, by M5Stack, Shenzhen, China). Data have been collected wirelessly using a Wi-Fi communication protocol on a Raspberry PI 4 (Raspberry Pi, Wales, UK). T and RH have been monitored using a commercial environmental sensor (BME280 by BOSCH, Gerlingen, Germany) connected via I2C to the M5Stick-C Plus.

### 2.3. Metrological Characterization of the Strain Sensors

To assess the use of the developed sensors for monitoring stem and fruit growth and their operation in an unstructured environment, they have been metrologically characterized to strain and RH. In particular, the sensors’ sensitivity to strain and the influence of RH on the sensors’ output were assessed.

#### 2.3.1. Stem Growth Sensor Characterization

The sensor’s sensitivity to strain has been tested using a testing machine (Instron 3365) by straining the sensor from 0% to 20% (i.e., from 0 mm to 12 mm) at a speed of 5 mm·min^−1^. This low speed was set to consider the trial quasi-static. The strain test has been performed five times. Figure 2a shows the picture of the experimental setup used for the metrological characterization. The sensor’s output was collected wirelessly using the electronics reported in Section 2.2.1. The sensor data along with the sensor strain have been recorded simultaneously. From the five trials performed, the average curve and its related expanded uncertainty have been calculated to enable the calculation of the sensor’s sensitivity to strain within the tested strain interval. The expanded uncertainty was calculated considering a *t*-Student distribution with four degrees of freedom since we performed five repeated trials and a confidence level of 95%. Since the sensor response is not linear, its sensitivity is not constant and cannot be defined with a single value. Therefore, we reported the average sensitivity to strain by considering the full-scale span to give a rough indication of this metrological parameter. The average sensitivity has been calculated from the relative resistance change ($\Delta R/R_{0}$) as in the following equation:(1)$$S=\frac{\frac{\Delta R}{R_{0}}{{|}_{\epsilon =20\%}-\frac{\Delta R}{R_{0}}|}_{\epsilon =0\%}}{20\%}$$

#### 2.3.2. Fruit Growth Sensor Characterization

The sensor’s sensitivity to strain has been assessed as detailed in the previous section. The only difference was in the maximum ϵ applied which was equal to 50%. This parameter has been set due to the application of interest in which the fruit exhibits a broader growth compared to the plant stem, thus a higher strain. A single trial has been performed to understand the sensor’s behavior within the strain range.

The sensor’s capability to measure the fruit growth was assessed by positioning the developed sensor around 3D-printed PLA discs of known diameter (ranging from 65 mm to 115 mm, with a step increase in diameter of 10 mm, and thickness of 20 mm) and waiting a sufficient time for the sensor to reach its regime (approximately 120 s). This diameter range has been chosen to easily cover typical values of melon dimensions from the first days of transplanting to harvesting. Then, the resistance, and the $\Delta R/R_{0}$ of the sensor were calculated. The sensor’s output was collected using the same setup as in Section 2.2.1.

To assess the sensor’s behavior when wrapped a fruit, the collected data have been processed following different steps. Firstly, the resistance and the $\Delta R/R_{0}$ have been calculated from the sensor’s output. Then, the last 30 s of the regime state of the sensor were used to calculate the mean value and its related expanded uncertainty. Figure 2b shows the printed 3D structure used to perform the trials, and the sensor’s positioning for the trials. Finally, from the mean values obtained, the average sensitivity has been calculated by linear fitting the experimental data.

#### 2.3.3. RH Influence Assessment

The influence of RH on the sensor’s output was assessed in the absence of strain by investigating the $\Delta R/R_{0}$ changes to different RH levels. This investigation was performed on both a bare and an encapsulated sensor to evaluate the effect of the encapsulation. Sensors were fixed on a flat surface at constant temperature (in the range 25.7 ± 0.5 °C considering the whole experiment) in a sealed box and two humidity tests were performed: (i) RH was varied from ∼10% to ∼98% and brought back down to ∼10% and (ii) RH was kept for over 3 h at ∼100%. Temperature and RH were continuously recorded by a thermistor (EL-USB-TP-LCD, LASCAR electronics, Whiteparish) and a capacitive humidity sensor (HIH 4000-002, Honeywell, Padova, Italy) simultaneously to the output of the tested sensors.

### 2.4. Experimental Tests

#### 2.4.1. Stem Growth in a Laboratory Assessment

The tobacco plant was positioned into an opened-top polycarbonate chamber to avoid any gusts of wind and allow T exchange. The plant light exposure was regulated by employing a wide spectrum LED light (PlantGrowLight-MAM-4H-01 by Maxsure) with 8 h of light exposure. The growth sensor was fixed on the plant’s stem (approximately 30 mm in length and 6 mm in width) using an elastic kinesiology tape (kinesiology tape by Alpidex, Toeging am Inn, Germany) and the microclimate monitoring electronics have been positioned into the plant’s vase. The sensor and environmental data have been collected at 10 Hz. To obtain a reference for the plant’s growth, 2 photo-reflective markers were cut into circles of 6 mm in diameter from a sheet of reflective material (3M^TM^ Scotchlite^TM^ Reflective Material 8910 Silver Fabric). The markers were positioned at the extremities of the sensor to monitor the same stem portion. To track the trajectories of the markers, an IR camera (Longruner LC26-US, with an image resolution of 2592 pixels × 1944 pixels) was used to capture images every 5 min and was positioned at a distance of 250 mm from the plant. The total monitoring time was of ∼3 days. Figure 3 shows a graphical representation of the experimental setup.

#### 2.4.2. Fruit Growth in an Open Field

The melon plant was placed in an open field exposed to direct sunlight for ∼10 h during the day. The growth sensor was positioned around the fruit at its center, and the compensating sensor and the microclimate platform were positioned on a rigid plastic tray elevated from the soil close to the growth sensor. Figure 4 shows a graphical representation of the experimental setup. As a reference for the melon growth, manual tape measurements of the fruit’s diameter were performed at the beginning and at the end of the trial. The total monitoring time was of ∼21 h. All data were collected at 10 Hz.

### 2.5. Data Analysis

For the stem growth sensor, the outputs of the two conductive sensors (i.e., ΔV1 and ΔV2) were firstly filtered using a 1st order low-pass filter with a cutoff frequency of 0.01 Hz. The choice of this low cutoff frequency can be considered adequate to monitor growth since it is a very slow phenomenon and the collected signal is characterized by frequencies lower than 0.01 Hz. To compensate the influence of T and RH from the stem sensor, the output of the compensating sensor (ΔV2) has been point-by-point subtracted from the stem sensor (ΔV1), thus obtaining the compensated sensor output (ΔV).

For the stem growth reference data, the collected images have been processed using a free video analysis and modeling tool built on the Open Source Physics Java framework (Tracker—[26]). For each reference marker, the position expressed in pixels has been identified along the *x*- and *y*-axes. Then, this position has been used to retrieve the total displacement by converting the pixel displacement into a spatial displacement (expressed in mm) using the first frame as calibration frame. Finally, the total stem growth has been calculated as the Euclidean distance between the two markers (ΔL).

For the fruit growth sensor, the output of the sensor ($\Delta V_{f}$) was firstly filtered using a moving average filter considering a time window equal to 10 min. This time window was chosen due to the growth phenomenon which occurs over several hours. Then, the obtained growth data have been fitted with a 1st order polynomial to retrieve information on the growth trend.

## 3. Results

In the present section, the results obtained from the data analysis of the stem and fruit growth sensors are presented.

### 3.1. Metrological Characterization of the Strain Sensors

#### 3.1.1. Stem Growth Sensor Characterization

The average sensitivity to strain showed by the sensor in the interval ϵ = 0–20% resulted in being −1.28. The negative *S* value is due to the resistance decreasing with strain, as shown in Figure 5. In addition, the decreasing monotonic trend showed by the sensor along with its sensitivity enables to assert that the sensor can be apt for monitoring the plant stem growth. Figure 5 shows the obtained average curve and its related uncertainty.

#### 3.1.2. Fruit Growth Sensor Characterization

The obtained sensitivity within the tested range (i.e., ϵ = 0–50%) resulted in being −1.04. The sensor’s output exhibited a monotonic decreasing trend (see Figure 6). The overall sensor’s output within the full range of operation shows a quadratic relationship with strain. However, the obtained values of sensitivity and the presence of a monotonic decreasing trend makes the sensor apt for monitoring strain.

The additional tests performed on the 3D-printed PLA discs confirmed the capability of the sensor to monitor changes in diameter thanks to the fixing method implemented. Indeed, as shown in Figure 7, a monotonic trend is exhibited with increasing diameters. The overall trend can be fitted with a second order polynomial and finds good agreement with both of the results obtained in Figure 5 and Figure 6.

### 3.2. RH Influence Assessment

The $\Delta R/R_{0}$ change in the sensors when exposed from 10% to 98% RH resulted in being 45% and 2% for the bare and encapsulated sensor, respectively. Moreover, under 100% RH for a long period of time (i.e., over 3 h), the $\Delta R/R_{0}$ change occurred was of 100% and 5% for the bare and encapsulated sensors, respectively. Figure 8 shows the $\Delta R/R_{0}$ change of the sensors under the tested conditions. In addition, in both of the tests, the bare sensor follows the RH dynamic, underlining how relevant its influence is on the sensor’s output.

### 3.3. Stem Growth in a Laboratory Assessment

The microclimate surrounding the plant exhibited broad variations throughout the experimental test according to the day/night cycle. Figure 9 shows T and RH trends throughout the test, in which the high peaks represent daytime and the valley represents nighttime when the building air conditioning is switched off.

The broad changes in T and RH underline how the compensating sensor employed in the experiment was pivotal for obtaining an accurate measurement of growth. The compensated output of the stem growth sensor (ΔV) is reported against time in Figure 10.

Finally, by plotting the sensor’s output versus the displacement calculated from the reference system, a good agreement was found and is testified by a direct relationship between the two. Over ∼3 days of monitoring the stem experienced a growth of ∼6.5 mm which corresponded to a voltage variation of ∼0.30 V. The increasing trend can be expressed with a linear fitting with a slope of 0.0535 V/mm. Figure 11 shows the output of the growth sensor against the reference system.

### 3.4. Fruit Growth in an Open Field

From the experimental data, the proposed sensor was capable of detecting the fruit growth. Indeed, a slight increasing trend has been found in the sensor’s output signal and can be seen in the positive slope of the linear fitting shown in Figure 12, which is 0.001731 V·h^−1^. Comparing this measurement with the manual tape measurements performed at the beginning and end of the trial good agreement is found. Indeed, the tape measurement showed a growth of ∼3 mm.

The wearable system was also able to simultaneously collect data on the microclimate surrounding the fruit has shown in Figure 13. The employed Wheatstone bridge in half-bridge configuration was able to compensate for the high temperature reached throughout the trial (i.e., maximum measured temperature in direct sunlight ∼58.4 °C) and in conjunction with the encapsulation of the sensors, the influence of RH was also compensated (i.e., maximum RH reached ∼76.7%).

## 4. Discussion

In the present work, we performed the feasibility assessment of a multi-sensor plant wearable platform for monitoring plant growth in terms of stem and fruit development and its surrounding microclimate in both a laboratory setting and an open field scenario.

In the literature, multisensory platforms have been proposed for monitoring plant health. These sensory devices have been commonly used to simultaneously monitor growth metrics and environmental parameters (e.g., RH, T, light intensity) that can affect essential processes occurring at the leaf level, such as transpiration and photosynthesis. In [5], the authors proposed the first portable sensor for microclimate monitoring able to detect RH and T changes. The sensor consists of a butterfly-shaped polymer matrix with a metal layer of Ti/Au for monitoring T and RH. Zhao and coauthors presented another multiplatform consisting of an ultrathin polymer layer encapsulating multiple sensing elements to measure T, light intensity, and growth [27]. Similarly, in [28], a stretchable structure with multiple sensing elements to monitor three environmental factors (RH, T, and light intensity) and leaf transpiration was presented. The system was designed in a cross shape but only the arm for transpiration monitoring was taped on the leaf lower epidermis while the others were suspended in the air. In the literature, the employment of sensors for simultaneously monitoring growth and microclimate has been carried out to increase the plant’s survival and optimize the crop’s product quality. All these solutions rely on printed and flexible sensors, which, in some cases, are characterized by complex manufacturing processes and the high cost of the read out electronics [5,9,10,16,18,29]. To cope with these concerns, we provided a facile and cost-effective solution to manufacture a strain sensor tailored to the plant needs and dimensions used to be integrated into a multi-sensor platform for continuously monitoring microclimate and growth. In addition, we pursued the encapsulation of the sensors for growth monitoring into silicone matrices to help in dampening the influence of climate variations on the sensor output. We also compensated for the T effects by using a dummy strain sensor on the adjacent arm in a Wheatstone Bridge configuration. In more detail, the sensing element devoted to measuring stem and fruit dimensional changes were directly mounted on the plant organs while the ones for T and RH monitoring in the plant’s surroundings. This approach helps in decoupling the sensor output changes induced by plant physiological processes from those caused by microclimate variations and make negligible the T effects on the strain sensors. The main strategies that we implemented to reduce the influence of microclimate on the growth sensors were the encapsulation of the strain sensors into silicone matrices to help in dampening the influence of the environment on their response and the integration of a dummy sensor in a Wheatstone Bridge configuration for T compensation. At the same time, we placed the sensing elements devoted to T and RH monitoring close to the engineered plants but not in direct contact with its surface similarly [28] to avoid any influence of plant physiological process on the measurement of environmental T and RH. However, deeper investigations should be carried out to make this impact negligible.

From this pilot study, the system has proven to be able to monitor the plant’s surrounding microclimate (i.e., T and RH) and the plant growth (i.e., stem and fruit growth). The stem monitoring wearable showed good agreement with the reference camera as testified by the almost monotonic trend obtained by plotting the sensor’s output against the reference data (Figure 11). Moreover, the bare fabric is highly influenced by RH as showed by the results obtained in Section 3.2 and Figure 8. Therefore, the compensation technique proposed by employing a nominally identical additional sensor resulted in being valid although it requires to build a more complex experimental setup with additional electronics. This complexity may not be sustainable in broad monitoring scenarios (e.g., crop fields) where the number of boards is already large.

Regarding this aspect, we also proposed a novel method for limiting the influence of RH on the sensor’s output and reducing the complexity of the experimental setup. For limiting the influence of RH on the sensor’s output, the sensor has been encapsulated in a three-layer structure composed by a silicone matrix. This multi-layer structured allowed for obtaining an easy manufacturing process which drastically improves the sensor’s resistance to RH. Moreover, from the electronics point of view, a different configuration of the Wheatstone bridge was proposed (i.e., half-bridge configuration) to reduce the number of boards to only one and improve the accuracy of measurements thanks to the additional sensor and the polymeric encapsulation.

To assess the reliability of this solution along with the sensor’s capability to measure fruit growth, a melon was provided with the wearable system and was monitored over a 21 h session in an open field with approximately 10 h of direct sunlight. The obtained results were promising. Indeed, the proposed encapsulation and electronic circuit for transduction were able to compensate for the broad changes in T (up to ∼58.4 °C) and RH (up to ∼76.7%).

The sensor was able to measure the fruit growth in agreement with the reference tape measurements and testified by the positive slope of the fitting polynomial. However, the sensor output exhibited a noisy output which might be related to the exposure to direct sunlight for a long period of time (∼10 h) since this noise was not present in the characterization trials performed in a laboratory setting.

## 5. Conclusions

In conclusion, this pilot study presents a cost-effective and reliable solution for monitoring plant growth and its surrounding microclimate in short-term monitoring sessions in both structured environments and open fields. The proposed solutions for compensating for the environmental parameters have proven to be a valuable option even when the system is employed in harsh conditions. Additional testing will be performed to test the proposed wearable on a broader field and with a higher number of boards to move towards a more realistic scenario in the framework of “precision agriculture” and “smart agriculture”. Moreover, improvements will be performed on the sensor’s manufacturing process to try to reduce, as much as possible, the noise on the sensor’s output signal probably related to the exposure to sunlight. Finally, we foresee integrating additional sensors (e.g., chemical sensors) to monitor other key parameters (e.g., pesticide concentration, soil properties) to further optimize the agronomic procedures in order to obtain a more versatile and all-around wearable system. To date, no plant wearables are on the market because some ongoing challenges with this technology should be overcome to ensure a solid on-organ adhesion, high compatibility, scalability, and reproducibility, with the lowest invasiveness, especially when used for long-term monitoring. Most of the commercially available wearable devices are systems for monitoring human health. However, recently, the use of wearables in the smart agriculture has been considered of pivotal relevance not only to provide insights into the performance of agricultural inputs for speeding up their commercial translation but also for assisting in evaluation of real-time plant development for optimizing their growth in a sustainable way. This will foster the introduction of these devices on the market. However, we know that further investigations are still needed to better assess the performance of our device and then improvements in scalability and long-lasting mechanical and chemical stability should be pursued to reach high measurement accuracy in real agricultural fields even under harsh environmental conditions. Despite these challenges, we expect that the next future of wearable sensors (including our platform) in the coming era of digital farms and precision agriculture will be bright.

## Figures and Tables

**Figure 1 sensors-23-00549-f001:**
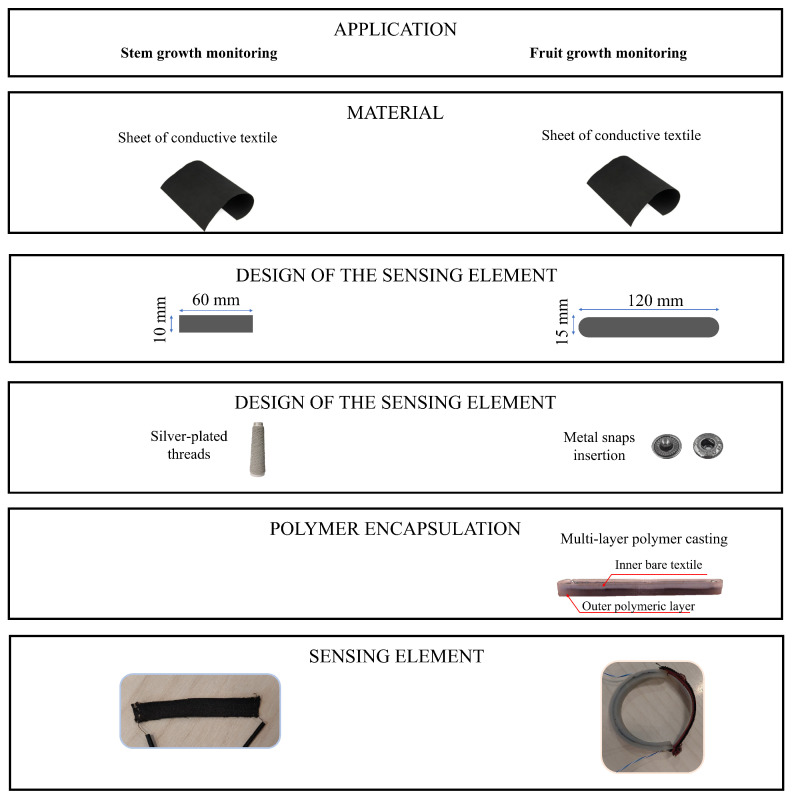
Graphical representation of the manufacturing process for the stem and fruit growth sensors.

**Figure 2 sensors-23-00549-f002:**
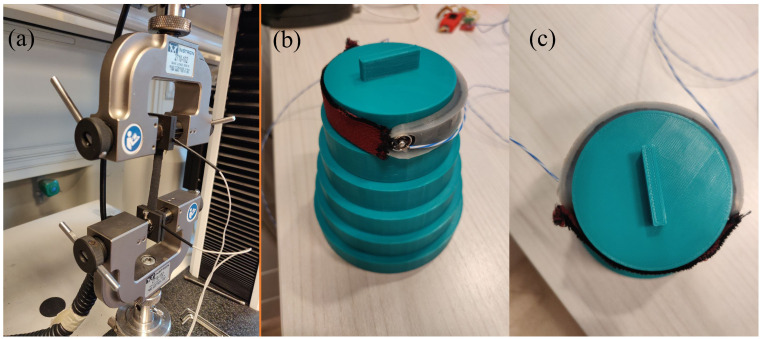
(**a**) Picture of the experimental setup for the metrological characterization of the stem growth sensor.In particular, the Instron clamps and the sensor positioning are shown; (**b**) side-view and (**c**) top-view of the 3D-printed discs used for assessing the fruit growth sensor’s behavior when simulating different fruit diameters at different growth stages (ranging from 65 mm to 115 mm). Additionally, the sensor’s placement for the trials is shown.

**Figure 3 sensors-23-00549-f003:**
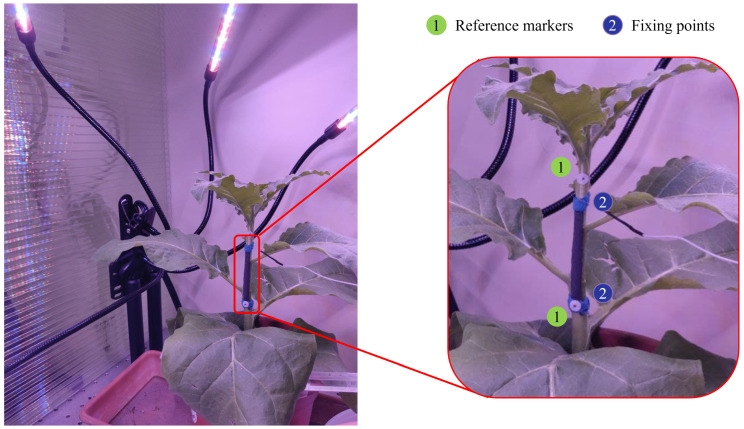
Experimental setup for the stem growth monitoring. The polycarbonate chamber in which the plant and the LED lights were placed is showed on the left. The sensor fixing points and the growth reference markers positioning on the stem are shown on the right.

**Figure 4 sensors-23-00549-f004:**
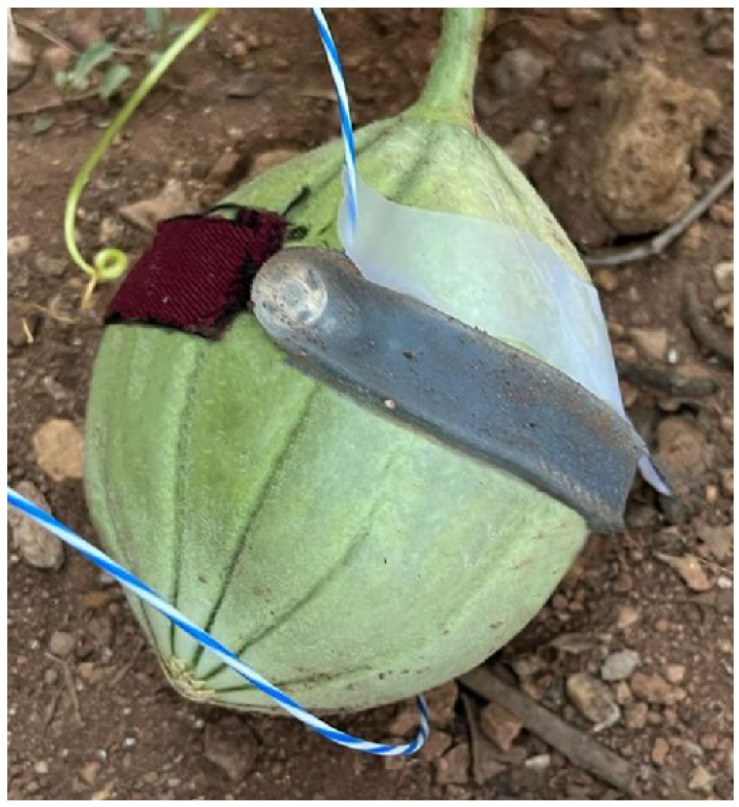
Graphical representation of the sensor positioning on the tested melon in the open field.

**Figure 5 sensors-23-00549-f005:**
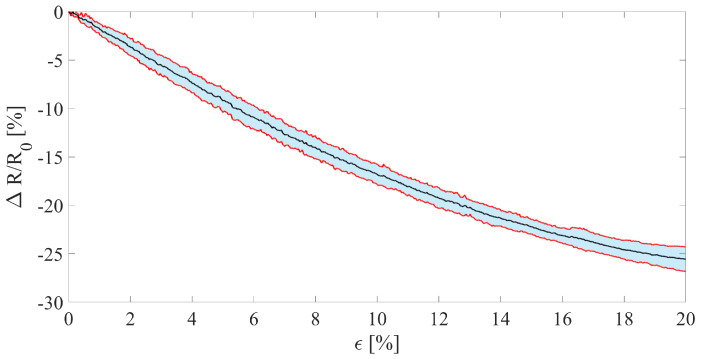
Relative resistance change of the stem growth sensor when strain up to 20%. The black line represents the average output curve, the red lines delimit the blue shaded area which represent the expanded uncertainty.

**Figure 6 sensors-23-00549-f006:**
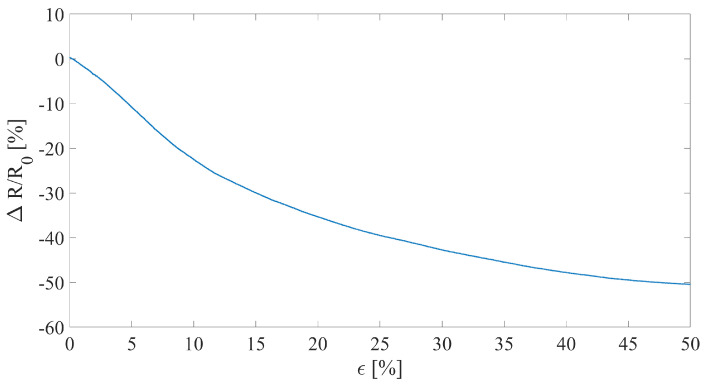
Relative resistance change of the fruit growth sensor when tested in a single strain test up to 50%.

**Figure 7 sensors-23-00549-f007:**
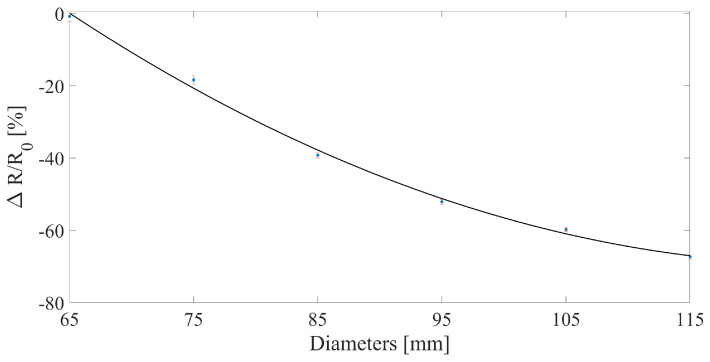
Relative resistance change of the fruit growth sensor when tested on 3D-printed discs ranging from 65 mm to 115 mm. The blue dots represent the mean value of the sensor’s relative resistance change over 30 s at regime, the red errorbars represent the expanded uncertainty, and the black line represents the 2nd order polynomial fitting of the experimental data.

**Figure 8 sensors-23-00549-f008:**
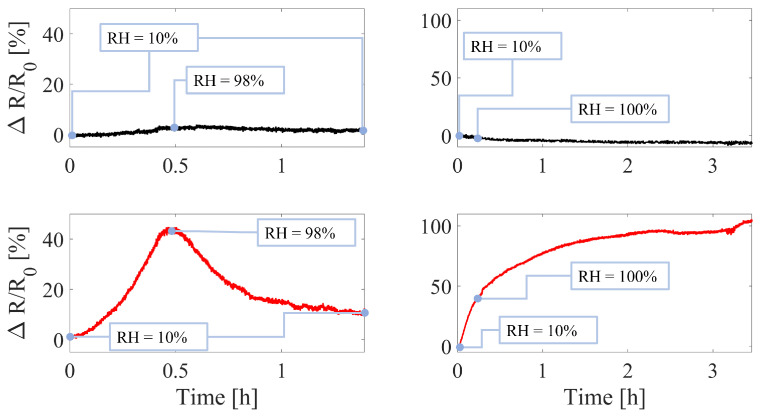
Relative resistance change of the two tested sensors undergoing RH changes. On the left side, the data collected by changing RH from ∼10% up to ∼98% and back to ∼10% are reported. The two trends on the right side show the relative resistance changes when the sensors are exposed at ∼100% RH for over 3 h (left). The black and red trends represent the relative resistance change of the encapsulated and bare sensors, respectively.

**Figure 9 sensors-23-00549-f009:**
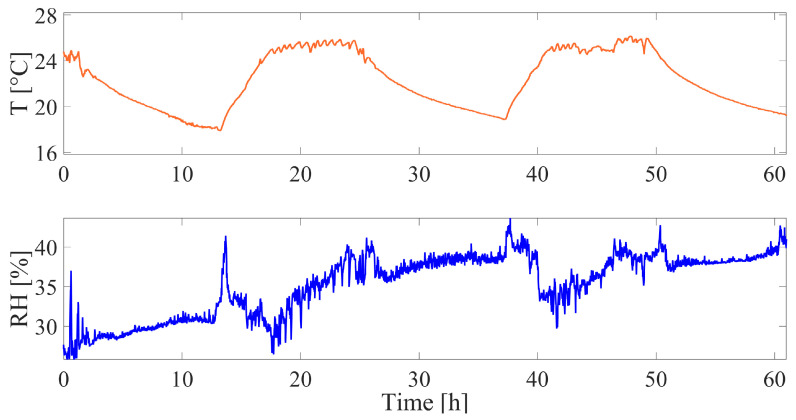
Microclimate data collected during the stem growth trial: temperature (T) trend versus time is showed in the top plot and relative humidity (RH) trend versus time is showed in the bottom plot during the entire experiment.

**Figure 10 sensors-23-00549-f010:**
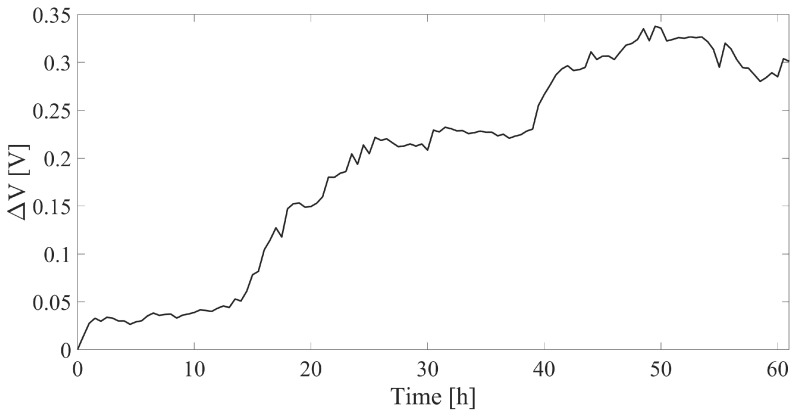
T- and RH-compensated output of the growth sensor plotted against time during the experimental trial.

**Figure 11 sensors-23-00549-f011:**
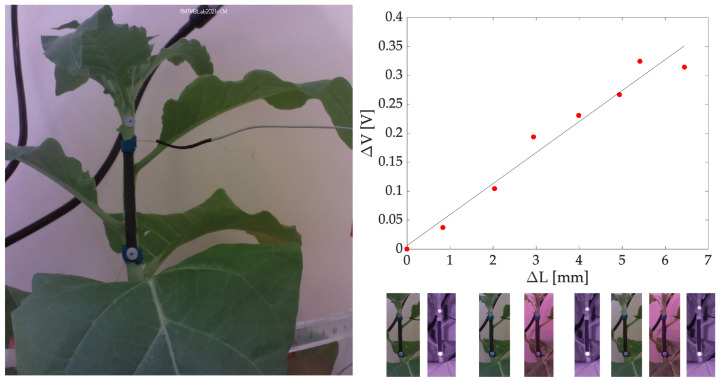
T and RH compensated output of the growth sensor plotted versus the stem growth measured by the reference camera. The best fitting line is also shown. Growth sample points have been collected every ∼8 h.

**Figure 12 sensors-23-00549-f012:**
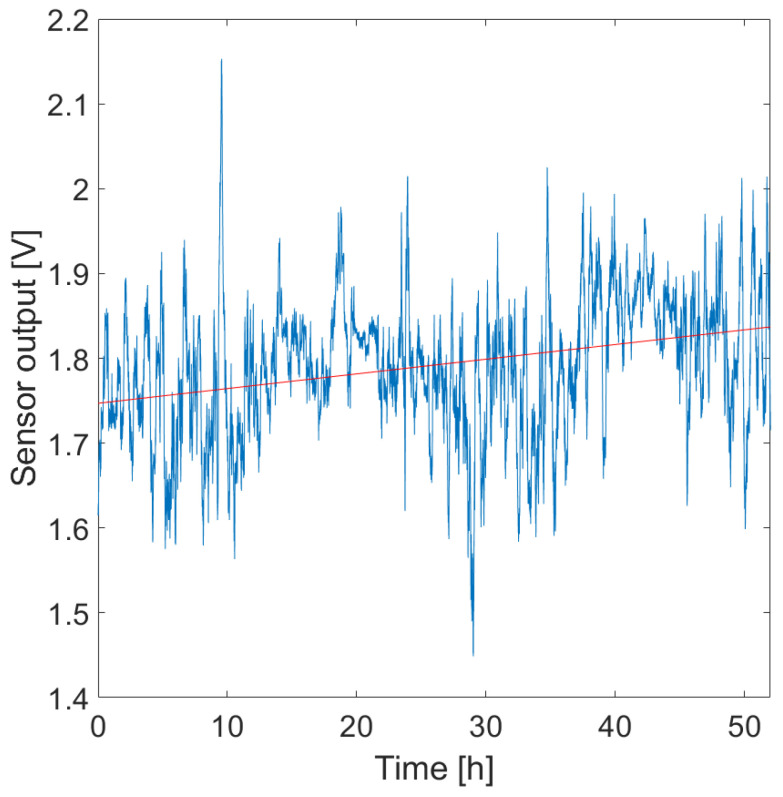
Sensor’s output plotted over time (blue line), and 1st order polynomial fitting showing the growth trend (red line).

**Figure 13 sensors-23-00549-f013:**
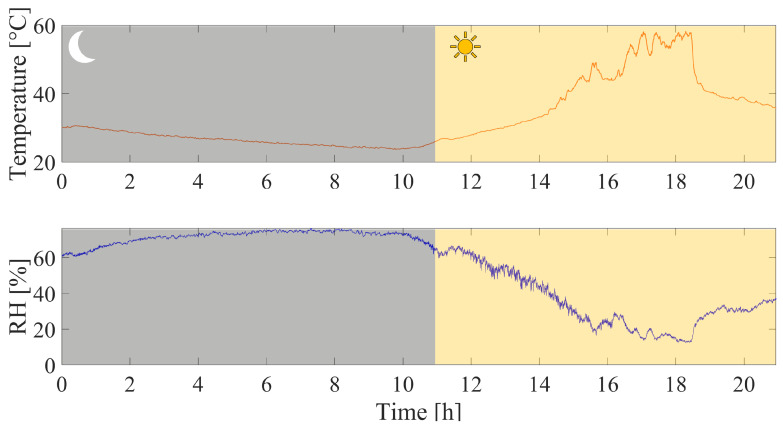
T (**top plot**) and RH (**bottom plot**) changes during the experimental trial performed on a melon in an open field.

## Data Availability

The data presented in this study are available on request from the corresponding author. The data are not publicly available due to privacy reasons.

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
