# Peer review of "Plant-Wear: A Multi-Sensor Plant Wearable Platform for Growth and Microclimate Monitoring"

_sensors, 2023, doi:10.3390/s23010549_

Round 1

Reviewer 1 Report

The authors report on a multi-sensor plant wearable platform for continuous and real-time monitoring of plant growth (stem, fruit) and microclimate (temperature, relative humidity). With their study, the authors want to find a cost-effective and reliable solution to increase the plant’s survival and optimize the crop’s product quality.

In particular, the authors design and carry out experiments to monitor, continuously and in real-time, the stem growth of a tobacco plant in a laboratory environment and the fruit growth of a melon plant in an open field along with their surrounding microclimate.

The work is well written. The Materials and Methods section is clear and detailed, the Experimental Tests are well described. However, to demonstrate the reliability of their system and produce robust results much longer measurements and much bigger data collection are required. The duration of the proposed experiments is considered insufficient to properly monitor the stem and fruit growth and understand the impact of the surrounding microclimate on their growth. I, therefore, recommend the authors to repeat their experiments for much longer time periods.

The comparisons with reference data and between laboratory and field measurements are incomplete, and the conclusions are not convincing.

Several aspects need to be clarified about the design of experiments and obtained results before considering the manuscript ready for acceptance.

In section 2.3.2, the authors use a 3D-printed disc for simulating different fruit diameters at different growth stages from 65 mm to 115 mm (50 mm span), but the size of the melon and the measured growth over time are not clearly specified. This does not allow for a direct comparison between the simulation and the real experiment.

Regarding the stem growth, the authors report about an overall growth of about 6.5 mm over a period of three days. Information about the size of the plant would help to quantify better the proportion of this growth.

The authors should explain better how they determine the sensitivity. A supporting reference to eq. 1 would be useful. Importantly, since there is not a linear dependence between data points in both Fig. 5 and Fig. 6, the sensitivity cannot be calculated as one value in the whole measurement range of 0-20% (Fig. 5) or 0-50% (Fig. 6), and expressions like “almost linear trend” (line 242, line 249) are not sufficient to justify the inappropriate approximation to linear dependence in the whole measurement range.

It is not clear how the uncertainty was calculated in Fig. 5. A brief explanation would help the reader to better understand. Similarly, adding a best fit curve and uncertainty in Fig. 6 is recommended.

In the caption of Fig. 8, the authors explain “RH changes from ca. 10% up to ca. 98% and back to ca. 10% (left)” but the indication about “back to ca. 10%)” seems to be missing in the figure. Also, in the expression “when exposed at ca. 100% RH for over 3h (left)”, the word “(left)” should be corrected with “(right)”.

The authors report about T and RH compensation but some further information would be needed to understand Fig. 10 and the overall reason to compensate for T and RH. Since T and RH are two important parameters that affect the plant growth, is it methodologically correct to compensate for them or, instead, the presence of such parameters should be considered and carefully studied?

Regarding Fig. 11, the authors should specify the time interval over which each data point was acquired. Also, a clarification about the meaning of the different colors of the pictures taken with the reference camera would help to understand, e.g., if different illuminations were used and how they affected the stem growth. Besides, the authors conclude that there is a good agreement between the strain sensor data and the reference camera (lines 309-311), but this statement should be better supported by a linear fit of the reported data, which should be included in the plot together with error bars.

The positive slope calculated in Fig. 12 is questionable due to the high noise level of the sensor output. To conclude about a positive slope, the authors are recommended to carry out much longer tests.

In conclusion, major revisions are required before consideration for publication in this journal.

Author Response

Response to reviewers

Plant-wear: A Multi-sensor Plant Wearable Platform for Growth and Microclimate Monitoring

Joshua Di Tocco, Daniela lo Presti*, Carlo Massaroni, Stefano Cinti, Sara Cimini, Laura de Gara, Emiliano Schena.

Sensors – MDPI

GENERAL CONSIDERATIONS

Primarily we would like to thank the Reviewers for their thorough revision and valuable observations, which helped us improve the earlier version of the paper. It is always extremely helpful to have an external opinion on sections that need clarification. We fundamentally agree with all the comments made by the Reviewers, and we have incorporated corresponding revisions into the revised version of the manuscript. The suggestions provided have been precious in the re-writing of the paper, as we hope it emerges from the examination of the present document. Below, it is possible to find our point-by-point response to reviewers’ comments. 

It is our feeling that along the line of reviewing the paper, we managed to improve it in both presentation and readability. 

Reviewer #1 (Remarks)

General comment. 

The authors report on a multi-sensor plant wearable platform for continuous and real-time monitoring of plant growth (stem, fruit) and microclimate (temperature, relative humidity). With their study, the authors want to find a cost-effective and reliable solution to increase the plant’s survival and optimize the crop’s product quality.

In particular, the authors design and carry out experiments to monitor, continuously and in real-time, the stem growth of a tobacco plant in a laboratory environment and the fruit growth of a melon plant in an open field along with their surrounding microclimate.

The work is well written. The Materials and Methods section is clear and detailed, the Experimental Tests are well described. However, to demonstrate the reliability of their system and produce robust results much longer measurements and much bigger data collection are required. The duration of the proposed experiments is considered insufficient to properly monitor the stem and fruit growth and understand the impact of the surrounding microclimate on their growth. I, therefore, recommend the authors to repeat their experiments for much longer time periods.

The comparisons with reference data and between laboratory and field measurements are incomplete, and the conclusions are not convincing.

Several aspects need to be clarified about the design of experiments and obtained results before considering the manuscript ready for acceptance.

Answer to general comment

We thank the referee for the positive feedback regarding the manuscript (well written and M&M clear and detailed) and for all the suggestions regarding the data collection (we have reported a longer analysis in figure 12 as recommended), we also detailed the comparison with reference and reorganized the conclusion of the articles. All these changes are highlighted in the reviewed version. In addition, we answered all the specific comments reported below.

Specific comments. 

Comment 1.1 In section 2.3.2, the authors use a 3D-printed disc for simulating different fruit diameters at different growth stages from 65 mm to 115 mm (50 mm span), but the size of the melon and the measured growth over time are not clearly specified. This does not allow for a direct comparison between the simulation and the real experiment.

Answer 1.1 We thank the reviewer for this sensible remark, that will help us clarify the motivation of this choice. We selected a range of diameters that can easily cover the ones of the melon from first weeks of transplanting to harvesting. To better motivate this choice we added the following sentence in section 2.3.2

“This diameter range has been chosen to easily cover typical values of melon dimensions from the first days of transplanting to harvesting.”

Comment 1.2 Regarding the stem growth, the authors report about an overall growth of about 6.5 mm over a period of three days. Information about the size of the plant would help to quantify better the proportion of this growth.

Answer 1.2 We thank the reviewer for this observation.

We installed the sensor for stem growth monitoring over a tobacco plant and we measured the growth starting from the initial distance between two points marked on the stem at the sensor level.  The overall size of the stem was approximately 30 mm in length and 6 mm in diameter. We added this information in the paper in Section 2.4.1.

Comment 1.3 The authors should explain better how they determine the sensitivity. A supporting reference to eq. 1 would be useful. Importantly, since there is not a linear dependence between data points in both Fig. 5 and Fig. 6, the sensitivity cannot be calculated as one value in the whole measurement range of 0-20% (Fig. 5) or 0-50% (Fig. 6), and expressions like “almost linear trend” (line 242, line 249) are not sufficient to justify the inappropriate approximation to linear dependence in the whole measurement range.

Answer 1.3 We want to thank the reviewer for raising this important aspect of the sensor response to strain. We completely agree that the sensitivity is not constant since the calibration curve is not linear. Therefore, in order to give an indication of the mean sensitivity in the whole range of measurement we calculated this value as the ratio between the output change and the input change considering the full-scale span (from 0% to 20%). However, we highlighted the fact that the sensitivity is not constant by changing section 2.3.1 as follows:

“Since the sensor response is not linear, its sensitivity is not constant and cannot be defined with a single value. Therefore, we reported the average sensitivity to strain by considering the full-scale span to give a rough indication of this metrological parameter. The average sensitivity has been calculated from the relative resistance change (DR/R0) as in the following equation”

As well as we deleted the awkward terms as “almost linear trend”.

Comment 1.4 It is not clear how the uncertainty was calculated in Fig. 5. A brief explanation would help the reader to better understand. Similarly, adding a best fit curve and uncertainty in Fig. 6 is recommended.

Answer 1.4 Thanks for this comment. To better explain how we evaluated the expanded uncertainly, we added the following part in

The expanded uncertainty was calculated considering a t-Student distribution with four degrees of freedom since we performed five repeated trials and a confidence level of 95%

Comment 1.5 In the caption of Fig. 8, the authors explain “RH changes from ca. 10% up to ca. 98% and back to ca. 10% (left)” but the indication about “back to ca. 10%)” seems to be missing in the figure. Also, in the expression “when exposed at ca. 100% RH for over 3h (left)”, the word “(left)” should be corrected with “(right)”.

Answer 1.5 We thank the reviewer for this wise advice which allows us to make clearer our figure. In order to accomplish this task, we replaced figure 8 with a new figure showing this indication. We also replaced the previous caption as recommended by the reviewer.

Comment 1.6 The authors report about T and RH compensation but some further information would be needed to understand Fig. 10 and the overall reason to compensate for T and RH. Since T and RH are two important parameters that affect the plant growth, is it methodologically correct to compensate for them or, instead, the presence of such parameters should be considered and carefully studied?

Answer 1.6 We really thank the reviewer for this observation.

When mounted on the plant surface, some physiological processes like transpiration and water transportation within the xylem (in the stem) may cause a change in plant local temperature and humidity and consequently, a not negligible influence on the DR variations of the sensor devoted to measure strain. Therefore, a compensation of these effects is needed to make more selective the growth sensor performance. Otherwise, to monitor environmental T and RH values, we introduced two sensing elements in the plant surrounding not in contact with the plant surface to decouple sensor output changes induced by plant physiological processes from those caused by microclimate variations.

We discussed this approach and the main issues related to these effects in section 3.3 just before the figure:

“The broad changes in T and RH underline how the compensating sensor employed in
the experiment was pivotal for obtaining an accurate measurement of growth.”

We also added this part in conclusion

“To cope with these concerns, we provided a facile and cost-effective solution to manufacture a strain sensor tailored to the plant needs and dimensions used to be integrated into a multi-sensor platform for continuously monitoring microclimate and growth. In addition, we pursued the encapsulation of the sensors for growth monitoring into silicone matrices to help in dampening the influence of climate variations on the sensor output. We also compensated for the T effects by using a dummy strain sensor on the adjacent arm in a Wheatstone Bridge configuration.”

Comment 1.7 Regarding Fig. 11, the authors should specify the time interval over which each data point was acquired. Also, a clarification about the meaning of the different colors of the pictures taken with the reference camera would help to understand, e.g., if different illuminations were used and how they affected the stem growth. Besides, the authors conclude that there is a good agreement between the strain sensor data and the reference camera (lines 309-311), but this statement should be better supported by a linear fit of the reported data, which should be included in the plot together with error bars.

Answer 1.7 We really thank the reviewer for this comment. We replaced the figure with the following that shows the linear trend. In addition, we reported the result of the linear fitting in section 3.3 as follows:

“Over ~3 days of monitoring the stem experienced a growth  of  ~6.5 mm which corresponded to a voltage variation of ~0.30 V. The increasing trend can be expressed with a linear fitting with a slope of 0.0535 V/mm”

Comment 1.8 The positive slope calculated in Fig. 12 is questionable due to the high noise level of the sensor output. To conclude about a positive slope, the authors are recommended to carry out much longer tests.

Answer 1.8 We really thank the referee for this advice. We replaced figure 12 with a new figure showing an analysis on a longer time slot (e.g., more than 2 days). Also, in this case the slope is positive and is equal to 0.001731 . Therefore, we replaced figure 12 and we also added this value of slope in the manuscript in Section 3.4.

Reviewer 2 Report

This manuscript proposed a multi-sensor plant wearable platform for monitoring the plant growth and microclimate parameters. The developed platform was tested in both a laboratory setting and an open field.

It is suggested that the authors should address the following issues for improving the quality of this manuscript.

First, it is suggested that the authors should re-summarize the abstract, in particular, the results. A common template of abstract includes background, methods, results, and conclusions. The authors only mentioned ‘The promising result’ in Line 11, however, one can hardly understand what a promising result implies.

Second, in the Introduction Section, several concepts were mentioned, including sustainable agriculture, precision agriculture, and smart agriculture. Although these concepts are very similar to some extent, they have minor differences. Consequently, the authors should keep consistent descriptions in the manuscript, otherwise, the authors have to clarify these concepts and further explain how their proposal could fulfil the current gap among the corresponding domain.

Third, In Lines 47-59, the authors introduced different technologies for monitoring the plant growth and microclimate parameters. However, as this manuscript focused on the wearable platform. It is suggested that the authors should provide a brief introduction about the state-of-the-art of wearable platforms in the domain of agriculture. And most importantly, what is the gap identified by the authors? Was the authors’ proposal able to fulfil this gap?

Fourth, In Section 2.1, the authors only introduced the materials of tobacco. The information about melons is missing.

Fifth, since this manuscript proposed a wearable platform, it is unclear the organization of this platform. At least, the overall picture of this platform should be given, indicating the main components, the communications between each component, etc. Besides, this is a multi-sensor platform. From the presentation of this manuscript, one can only understand there are several sensors being used. However, what sensors and how many sensors were used in the laboratory setting or the open field setting?

Sixth, the figures presented in this manuscript have low readability. For instance, in Figure 1, did the authors expect that one can replicate such a sensor from this graphical representation? What did the blue line imply, as well as the ‘add’ operation? Also, in Figure 2, markers should appear in each sub figures, it is totally unclear the meaning of each sub figure.

Seventh, the experiment lacks a comparative analysis with other work.

In general, the topic of this manuscript is interesting, however, plenty of technical details are missing, in particular, the overall framework of the proposed platform, resulting in difficulty on evaluating the experiment. 

Author Response

Response to reviewer

Plant-wear: A Multi-sensor Plant Wearable Platform for Growth and Microclimate Monitoring

Joshua Di Tocco, Daniela lo Presti*, Carlo Massaroni, Stefano Cinti, Sara Cimini, Laura de Gara, Emiliano Schena.

Sensors – MDPI

General comment. 

This manuscript proposed a multi-sensor plant wearable platform for monitoring the plant growth and microclimate parameters. The developed platform was tested in both a laboratory setting and an open field.

It is suggested that the authors should address the following issues for improving the quality of this manuscript.

In general, the topic of this manuscript is interesting, however, plenty of technical details are missing, in particular, the overall framework of the proposed platform, resulting in difficulty on evaluating the experiment.

Answer to general comment

We thank the referee for the positive feedback regarding the manuscript topic and for the wise revisions/ observations. We answered to all the specific comments reported below.

Specific comments. 

Comment 2.1 First, it is suggested that the authors should re-summarize the abstract, in particular, the results. A common template of abstract includes background, methods, results, and conclusions. The authors only mentioned ‘The promising result’ in Line 11, however, one can hardly understand what a promising result implies.

Answer 2.1 We thank the reviewer for this comment. We have reorganized the abstract in a structured way but without headings as suggested in Sensors MDPI template. The new version is the following:

Wearable devices are widely spreading in various scenarios for monitoring different parameters related to humans and recently plants health. In the context of precision agriculture, wearables have proven to be a valuable alternative to traditional measurement methods for quantitatively monitoring plant development. This study proposed a multi-sensor wearable platform for monitoring the growth of plant organs (i.e., stem and fruit) and microclimate (i.e., environmental temperature - T and relative humidity - RH). The platform consists of a custom flexible strain sensor for monitoring growth when mounted on a plant and a commercial sensing unit for monitoring T and RH values of the plant surrounding. A different shape was conferred to the strain sensor according to the plant organs to be engineered. A dumbbell shape was chosen for the stem while a ring shape for the fruit. A metrological characterization was carried out to investigate the strain sensitivity of the proposed flexible sensors and then, preliminary tests were performed in both indoor and outdoor scenarios to assess the platform performance. The promising results suggest that the proposed system can be considered one of the first attempts to design wearable and portable systems tailored to the specific plant organ with the potential to be used for future applications in the coming era of digital farms and precision agriculture”

Comment 2.2 Second, in the Introduction Section, several concepts were mentioned, including sustainable agriculture, precision agriculture, and smart agriculture. Although these concepts are very similar to some extent, they have minor differences. Consequently, the authors should keep consistent descriptions in the manuscript, otherwise, the authors have to clarify these concepts and further explain how their proposal could fulfil the current gap among the corresponding domain?

Answer 2.2 We thank the reviewer for this comment. We added this part in introduction.

“Precision agriculture is a modern farming management concept that uses digital techniques to adjust and fine-tune land for optimizing agricultural production processes. Here, the key point is the management optimization. Smart agriculture is a more recent concept that investigates the use of innovative technology to improve agricultural production while at the same time lowering the inputs significantly. Here, the focus is rather on access to data and the application of these data. Hence, smart agriculture runs on the principles of precision agriculture shifting to a holistic and more rounded approach where the focus is not only on management optimization but on the employment of smartest treatments. The use of these quantitative data to apply measures that are economically and ecologically meaningful makes precision agriculture sustainable.”

Comment 2.3 Third, In Lines 47-59, the authors introduced different technologies for monitoring the plant growth and microclimate parameters. However, as this manuscript focused on the wearable platform. It is suggested that the authors should provide a brief introduction about the state-of-the-art of wearable platforms in the domain of agriculture. And most importantly, what is the gap identified by the authors? Was the authors’ proposal able to fulfil this gap?

Answer 2.3 We thank the reviewer for this comment that help us to improve the quality of the paper. We added the following part in Conclusion:

To cope with these concerns, we provided a facile and cost-effective solution to manufacture a strain sensor tailored to the plant needs and dimensions used to be integrated into a multi-sensor platform for continuously monitoring microclimate and growth. In addition, we pursued the encapsulation of the sensors for growth monitoring into silicone matrices to help in dampening the influence of climate variations on the sensor output. We also compensated for the T effects by using a dummy strain sensor on the adjacent arm in a Wheatstone Bridge configuration.”

Comment 2.4 Fourth, In Section 2.1, the authors only introduced the materials of tobacco. The information about melons is missing.

Answer 2.4 We thank the referee for this advice and we replaced the previous version of 2.1 as follows:

2.1. Plant Material

The proposed platform has been tested on two different plants: i) in laboratory settings, a Tobacco plant (Nicotiana tabacum) was used to focus on stem growth. In this controlled experiment we sterilized the tobacco seeds by soaking in 70% ethanol for 2 min, followed by soaking in 5% bleach for 15 min. The seeds were then rinsed five times with sterilized water. Sterilized tobacco seeds were sowed in soil at 25 ± 1°C with a photoperiod of 16 h light/8 h dark; ii) in open field scenario, a melon plant (Cucumis melo) was used to assess the feasibility of the proposed sensor for monitoring the fruit growth.

Comment 2.5 Fifth, since this manuscript proposed a wearable platform, it is unclear the organization of this platform. At least, the overall picture of this platform should be given, indicating the main components, the communications between each component, etc. Besides, this is a multi-sensor platform. From the presentation of this manuscript, one can only understand there are several sensors being used. However, what sensors and how many sensors were used in the laboratory setting or the open field setting?

Answer 2.5 We thank the reviewer. We better explain the platform in Section 2.2.

“The proposed sensors (for stem and fruit growth) represent the custom elements of the multi-sensor platform consisting of strain sensors for monitoring stem elongation and fruit expansion and a commercial environmental system (BME280 by BOSCH, Gerlingen, Germany) for measuring environmental T and RH”

Comment 2.6 Sixth, the figures presented in this manuscript have low readability. For instance, in Figure 1, did the authors expect that one can replicate such a sensor from this graphical representation? What did the blue line imply, as well as the ‘add’ operation? Also, in Figure 2, markers should appear in each sub figures, it is totally unclear the meaning of each sub figure.

Answer 2.6 We thank for this comment. We updated Figure 1 and We replaced Figure 2, according to these observations.

Comment 2.7 Seventh, the experiment lacks a comparative analysis with other work.

Answer 2.7 We thank the reviewer for this observation.

We added the following part in Discussion:

In the literature, multisensory platforms have been proposed for monitoring plant health. These sensory devices have been commonly used to simultaneously monitor growth metrics and environmental parameters (e.g., RH, T, light intensity) that can affect essential processes occurring at the leaf level, such as transpiration and photosynthesis. In [3] authors proposed the first portable sensor for microclimate monitoring able to detect RH and T changes. The sensor consists of butterfly-shaped polymer matrix with a metal layer of Ti/Au for monitoring T and RH. Zhao and coauthors presented another multiplatform consisting of an ultrathin polymer layer encapsulating multiple sensing elements to measure T, light intensity, and growth [27]. Similarly, in [28], a stretchable structure with multiple sensing elements to monitor three environmental factors (RH, T, and light intensity) and leaf transpiration was presented. The system was designed in a cross shape but only the arm for transpiration monitoring was taped on the leaf lower epidermis while the others suspended in the air…

…In our study, we have also proposed a multisensory platform for growth and microclimate monitoring. In more details, the sensing element devoted to measure stem and fruit dimensional changes were directly mounted on the plant organs while the ones for T and RH monitoring in the plant surrounding. This approach helps in decoupling the sensor output changes induced by plant physiological processes from those caused by microclimate variations and make negligible the T effects on the strain sensors. The main strategies that we implemented to reduce the influence of microclimate on the growth sensors were the encapsulation of the strain sensors into silicone matrices to help in dampening the influence of the environment on their response and the integration of a dummy sensor in a Wheatstone Bridge configuration for T compensation. At the same time, we placed the sensing elements devoted to T and RH monitoring close to the engineered plants but no in direct contact with its surface similarly [28] to avoid any influence of plant physiological process on the measurement of environmental T and RH. However, deeper investigations should be carried out to make this impact negligible.”

Reviewer 3 Report

Current manuscript entitled “Plant-wear: A Multi-sensor PlantWearable Platform for Growth and Microclimate Monitoring” by “Tocco et al” provided a multi-sensor wearable platform for monitoring the plant surrounding microclimate (i.e., temperature and relative humidity) and the growth of two plant organs (i.e., plant stem and fruit). The platform was tested for monitoring the stem growth on a tobacco plant in a laboratory setting and for monitoring fruit growth on a melon in an open field. The promising results underline the potentiality of employing the platform for monitoring plant and fruit growth along with the plant’s microclimate. The proposed platform enables to obtain plant- and fruit-tailored sensors to be customized to the application of interest and encourages future testing. The authors choose an emerging topic. The work is interesting and written well and can be accepted after addressing the following comments.

1.      The authors choose an emerging and significant topic.

2.      In the introduction section authors mentioned about “different technologies have been employed for monitoring plant growth based on imaging technology and strain sensors.” (Lines 47-54). The authors should mention their advantages and disadvantages

3.      The authors should comment on the commercialization of the developed sensor.

4.      Recently portable devices have been developed for the detection of food contaminants. The authors should discuss about them.

https://doi.org/10.1016/j.ccr.2021.214305

https://doi.org/10.1016/j.bios.2020.112636

Author Response

Response to reviewer

Plant-wear: A Multi-sensor Plant Wearable Platform for Growth and Microclimate Monitoring

Joshua Di Tocco, Daniela lo Presti*, Carlo Massaroni, Stefano Cinti, Sara Cimini, Laura de Gara, Emiliano Schena.

Sensors – MDPI

Reviewer #3 (Remarks) 

General comment. 

Current manuscript entitled “Plant-wear: A Multi-sensor PlantWearable Platform for Growth and Microclimate Monitoring” by “Tocco et al” provided a multi-sensor wearable platform for monitoring the plant surrounding microclimate (i.e., temperature and relative humidity) and the growth of two plant organs (i.e., plant stem and fruit). The platform was tested for monitoring the stem growth on a tobacco plant in a laboratory setting and for monitoring fruit growth on a melon in an open field. The promising results underline the potentiality of employing the platform for monitoring plant and fruit growth along with the plant’s microclimate. The proposed platform enables to obtain plant- and fruit-tailored sensors to be customized to the application of interest and encourages future testing. The authors choose an emerging topic. The work is interesting and written well and can be accepted after addressing the following comments.

Answer to general comment

We thank the referee for the positive feedback regarding the manuscript and for the wise revisions/ observations. We answered to all the specific comments reported below.

Specific comments. 

Comment 3.1 The authors choose an emerging and significant topic.

Answer 3.1 We thank the reviewer for this positive feedback.

Comment 3.2 In the introduction section authors mentioned about “different technologies have been employed for monitoring plant growth based on imaging technology and strain sensors.” (Lines 47-54). The authors should mention their advantages and disadvantages.

Answer 3.2 We thank the reviewer for this useful.  

We added the following parts in Introduction:

“To date, traditional technologies employed for plant health monitoring include contactless methods like spectroscopy, machine vision systems, imaging techniques, and drones. [11-14] The use of remote systems faces some partial issues that are dampening their application in long-term plant monitoring. The lack of high spatial and temporal resolution and the low measurement reliability associated with these methods make them inadequate for the continuous tracking of plant organs development. Recently, new techniques have emerged to monitoring plant growth. Wearable systems embedding flexible strain sensors are reaching growing attention to overcome these issues thanks to their high stretchability and adaptability to plant organs. Most are conductive materials integrated within a polymer substrate or directly brushed on the plant surface with different principle of work [3,9,15,16]. Usually, a change in the electrical resistance or capacitance is experienced by the proposed sensing element under the growth of the engineered plant organs (e.g., stem, leaves, or fruit). Recently, fiber optic sensors (i.e., fiber Bragg gratings) have been proposed for the same aims with interesting results [8,17]”.

Comment 3.3 The authors should comment on the commercialization of the developed sensor.

Answer 3.3 We thank the reviewer for this comment in Conclusion.

“To date, no plant wearables are on the market because some ongoing challenges with this technology should be overcome to ensure a solid on-organ adhesion, high compatibility, scalability, and reproducibility, with the lowest invasiveness, especially when used for long-term monitoring. Most of the commercially available wearable devices are systems for monitoring human health. However, recently, the use of wearables in the smart agriculture has been considered of pivotal relevance not only to provide insights into the performance of agricultural inputs for speeding up their commercial translation but also for assisting in evaluation of real-time plant development for optimizing their growth in a sustainable way. This will foster the introduction of these devices on the market. However, we know that further investigations are still needed to better assess the performance of our device and then, improvements in scalability, long-lasting mechanical and chemical stability should be pursued to reach high measurement accuracy in real agricultural fields even under harsh environmental conditions. Despite these challenges, we expect that the next future of wearable sensors (including our platform) in the coming era of digital farms and precision agriculture will be bright”.

Comment 3.4 Recently portable devices have been developed for the detection of food contaminants. The authors should discuss about them.

https://doi.org/10.1016/j.ccr.2021.214305

https://doi.org/10.1016/j.bios.2020.112636

Answer 3.4 We thank the reviewer for this useful observation. We added the suggested studies in Introduction.

“In the context of sustainable agriculture, the optimization of the harvest may be supported by monitoring and quantifying plant growth and the influence of stressor factors. Plant stresses can be categorized into two classes: abiotic and biotic factors. Abiotic factors are related to the microclimate and include as an example, temperature level, irradiance, water availability, salinity, atmospheric carbon dioxide (CO2) enrichment, while biotic factors refer to damage caused by pest and pathogens [2-4]

Reviewer 4 Report

Journal Name: Sensors

Title: Plant-wear: A Multi-sensor Plant Wearable Platform for Growth and Microclimate Monitoring

In the current research, D.L. Presti et al. have made good research on plant wearable platforms for growth and microclimate monitoring. It is very necessary to develop wearable platforms for plants to monitor the various parameters of agricultural plants to develop the culture of smart culture. In this regard, the current work of the authors is very important and suits the MDPI sensors. But the article needs some major revisions to get published in MDPI sensors. 

1.      I feel the English of the article needs to be improved. Please recheck grammar and syntax errors.

2.      At the abstract mention discuss the results and important findings further. (Melon plant details not given)

3.      Can authors provide details of the stem growing up to 240 hours at an interval of 10?

4.      Significance of Fig 5 has to be discussed further.

5.      Please discuss the RH influence in a more elaborated way it is very difficult to understand.

6.      Discussion should be made in the results section

7.      Future perspectives should be added in detail in the conclusion part.

8.      MDPI template should be followed

9.      Recheck reference formats

Author Response

Response to reviewer

Plant-wear: A Multi-sensor Plant Wearable Platform for Growth and Microclimate Monitoring

Joshua Di Tocco, Daniela lo Presti*, Carlo Massaroni, Stefano Cinti, Sara Cimini, Laura de Gara, Emiliano Schena.

Sensors – MDPI

Reviewer #4 (Remarks) 

General comment. 

In the current research, D.L. Presti et al. have made good research on plant wearable platforms for growth and microclimate monitoring. It is very necessary to develop wearable platforms for plants to monitor the various parameters of agricultural plants to develop the culture of smart culture. In this regard, the current work of the authors is very important and suits the MDPI sensors. But the article needs some major revisions to get published in MDPI sensors

Answer to general comment

We thank the referee for the positive feedback regarding the manuscript and for the wise revisions/ observations. We answered to all the specific comments reported below.

Specific comments. 

Comment 4.1 I feel the English of the article needs to be improved. Please recheck grammar and syntax errors.

Answer 4.1 We thank the reviewer for this comment e we changed several parts through the whole manuscript. All these parts are highlighted.

Comment 4.2 At the abstract mention discuss the results and important findings further. (Melon plant details not given).

Answer 4.2 We thank the reviewer for this useful. We reorganized the abstract to accomplish this comment as follows:

Wearable devices are widely spreading in various scenarios for monitoring different parameters related to humans and recently plants health. In the context of precision agriculture, wearables have proven to be a valuable alternative to traditional measurement methods for quantitatively monitoring plant development. This study proposed a multi-sensor wearable platform for monitoring the growth of plant organs (i.e., stem and fruit) and microclimate (i.e., environmental temperature - T and relative humidity - RH). The platform consists of a custom flexible strain sensor for monitoring growth when mounted on a plant and a commercial sensing unit for monitoring T and RH values of the plant surrounding. A different shape was conferred to the strain sensor according to the plant organs to be engineered. A dumbbell shape was chosen for the stem while a ring shape for the fruit. A metrological characterization was carried out to investigate the strain sensitivity of the proposed flexible sensors and then, preliminary tests were performed in both indoor and outdoor scenarios to assess the platform performance. The promising results suggest that the proposed system can be considered one of the first attempts to design wearable and portable systems tailored to the specific plant organ with the potential to be used for future applications in the coming era of digital farms and precision agriculture.”

Comment 4.3 Can authors provide details of the stem growing up to 240 hours at an interval of 10?

Answer 4.3 We thank the reviewer for this useful observation. We highlighted the stem growing each 8 h with the corresponding change in the output voltage of the sensor. This information Is reported in figure 11. We highlighted this aspect in the caption of figure 11.

Comment 4.4 Significance of Fig 5 has to be discussed further

Answer 4.4 We thank the reviewer for this useful. We described better the Figure 5 and we added more information about the uncertainty evaluation in Section 2.3.1:

The expanded uncertainty was calculated considering a t-Student distribution with four degrees of freedom since we performed five repeated trials and a confidence level of 95%”

Comment 4.5 Please discuss the RH influence in a more elaborated way it is very difficult to understand

Answer 4.5 We thank the reviewer for this advice. In order to better describe RH influence, we replaced figure 8 and replaced the previous caption.

Comment 4.6 Discussion should be made in the results section

Answer 4.6 We thank the reviewer for this useful. We split “Discussions and Conclusion” by adding a new section Discussion to better compare with other studies in the literature as suggested by reviewers. Then, we also added new parts in Results to better discuss our achievements and we modify Figure 11 to better emphasize the linear trend of the sensor response.

Comment 4.7 Future perspectives should be added in detail in the conclusion part

Answer 4.7 We thank the reviewer for comment that help us to improve the quality of the manuscript. We added the following sentences in conclusion:

“Additional testing will be performed to test the proposed wearable on a broader field and with a higher number of boards to move towards a more realistic scenario in the framework of "precision agriculture" and "smart agriculture". Moreover, improvements will be performed to the sensor's manufacturing process to try to reduce, as much as possible, the noise on the sensor's output signal probably related to the exposure to sunlight. Finally, we foresee integrating additional sensors (e.g., chemical sensors) to monitor other key parameters (e.g., pesticide concentration, soil properties) to further optimize the agronomic procedures in order to obtain a more versatile and all-around wearable system. To date, no plant wearables are on the market because some ongoing challenges with this technology should be overcome to ensure a solid on-organ adhesion, high compatibility, scalability, and reproducibility, with the lowest invasiveness, especially when used for long-term monitoring. Most of the commercially available wearable devices are systems for monitoring human health. However, recently, the use of wearables in the smart agriculture has been considered of pivotal relevance not only to provide insights into the performance of agricultural inputs for speeding up their commercial translation but also for assisting in evaluation of real-time plant development for optimizing their growth in a sustainable way. This will foster the introduction of these devices on the market. However, we know that further investigations are still needed to better assess the performance of our device and then, improvements in scalability, long-lasting mechanical and chemical stability should be pursued to reach high measurement accuracy in real agricultural fields even under harsh environmental conditions. Despite these challenges, we expect that the next future of wearable sensors (including our platform) in the coming era of digital farms and precision agriculture will be bright.

Comment 4.8 MDPI template should be followed

Answer 4.8 We thank the reviewer for this useful. We updated the template.

Comment 4.9 Recheck reference formats

Answer 4.9 We thank the reviewer for this useful. We checked the format, and we edited the ones that differ from Sensors MDPI format. Then, we added Sensors as a Journal on the LaTeX template.

Round 2

Reviewer 2 Report

The authors have addressed my previous concerns. 

I have no further comments.

Reviewer 4 Report

The authors made sufficient changes article can be accepted in present form.